# A realist review of advance care planning for people with multiple sclerosis and their families

Laura Cottrell[1], Guillaume Economos[1], Catherine Evans[1,2‡], Eli Silber[3‡], Rachel Burman[1‡], Richard Nicholas[4‡], Bobbie Farsides[5‡], Stephen Ashford[1,6,7‡], Jonathan Simon Koffman[1]*

1 Florence Nightingale Faculty of Nursing, Midwifery & Palliative Care, King's College London, Cicely Saunders Institute, London, United Kingdom, 2 Sussex Community NHS Foundation Trust, Brighton General Hospital, Brighton, United Kingdom, 3 Department of Neurology, King's College Hospital NHS Foundation Trust, London, United Kingdom, 4 Department of Cellular & Molecular Neuroscience, Imperial College, Charing Cross Hospital, London, United Kingdom, 5 Brighton and Sussex Medical School University of Sussex, Brighton, United Kingdom, 6 UK & Regional Hyper-acute Rehabilitation Unit, London North West University Healthcare NHS Trust, London, United Kingdom, 7 Centre for Nursing, Midwifery and Allied Health Research, University College London Hospitals, London, United Kingdom

☉ These authors contributed equally to this work.
‡ These authors also contributed equally to this work.
* jonathan.koffman@kcl.ac.uk

**Data Availability Statement:** All relevant data are within the manuscript and its Supporting Information files (Appendix A & B, S1 File).

## Abstract

### Background

Advance care planning (ACP) is reported to improve the quality of outcomes of care among those with life-limiting conditions. However, uptake is low among people living with multiple sclerosis (MS) and little is known about why or how people with MS engage in this process of decision-making.

### Aims

To develop and refine an initial theory on engagement in ACP for people with MS and to identify ways to improve its uptake for those who desire it.

### Methods

Realist review following published protocol and reporting following Realist and Meta-narrative Evidence Synthesis: Evolving Standards (RAMESES) guidelines. A multi-disciplinary team searched MEDLINE, PsychInfo, CINAHL, Scopus, Web of Science, Embase, Google Scholar in addition to other sources from inception to August 2019. Quantitative or qualitative studies, case reports, and opinion or discussion articles related to ACP and/or end of life discussions in the context of MS were included, as well as one article on physical disability and one on motor neuron disease, that contributed important contextual information. Researchers independently screened abstracts and extracted data from full-text articles. Using abductive and retroductive analysis, each article was examined for evidence to support or refute 'context, mechanism, and outcome' (CMO) hypotheses, using the Integrated Behaviour Model to guide theory development. Quality was assessed according to

**Funding:** This review is part of a larger study that was generously funded by the MS Society [Grant code 93]. CE is funded by HEE/NIHR Senior Clinical Lectureship (ICA-SCL-2015-01-001).

**Competing interests:** The authors have declared that no competing interests exist.

methodological rigour and relevance of evidence. Those studies providing rich descriptions were synthesised using a realist matrix to identify commonalities across CMO configurations.

## Results

Of the 4,034 articles identified, 33 articles were included in the synthesis that supported six CMO hypotheses that identified contexts and mechanisms underpinning engagement in ACP for people with MS and included: acceptance of their situation, prior experiences, confidence, empowerment, fear (of being a burden, of death and of dying) and the desire for autonomy. Acceptance of self as a person with a life-limiting illness was imperative as it enabled people with MS to see ACP as pertinent to them. We identified the context of MS— its long, uncertain disease trajectory with periods of stability punctuated by crisis—inhibited triggering of mechanisms. Similarly, the absence of skills and confidence in advanced communication skills among health professionals prevented possibilities for ACP discussions taking place.

## Conclusion

Although mechanisms are inhibited by the context of MS, health professionals can facilitate greater uptake of ACP among those people with MS who want it by developing their skills in communication, building trusting relationships, sharing accurate prognostic information and sensitively discussing death and dying.

## Introduction

People with multiple sclerosis (MS) face an uncertain future. While some have little disability others can have profound physical and psychological limitations [1]. Approximately 40–70% of people with MS develop cognitive impairment throughout their illness experience, some severe [1–6]. Consequently, people with MS may have reduced decision-making ability, impairing their everyday functioning [7] that may limit their ability to plan for future circumstances.

Advance care planning (ACP) has been defined as a "process that supports adults at any age or stage of health in understanding and sharing their personal values, life goals, and preferences regarding future care" [8] and evolved from the advance directive movement for individuals to maintain control and plan their future care in the event of physical or mental incapacity [9]. Whilst advance directives consist of written directions, often within narrow and clearly marked boundaries (such as a Do Not Attempt Resuscitation [DNAR] order), ACP is broader in scope, linked to benefits for patients, health providers, and health systems [10]. The scope of ACP is appealing as it embodies a person and family-centred, holistic approach to facilitate decision-making. Importantly, ACP is not a one-time exercise but is a heterogeneous process of multiple discussions involving different actors across many care settings.

There is a growing interest among people with MS who want to talk about their future with health professionals [1, 11]. However, despite expressed concerns about end of life discussions and choices [12], few people with MS engage in these important conversations [13–15]. Findings from a 2016 study indicated that amongst neurology patients, people with MS are the least likely to be referred to palliative care services and the most likely to die in hospital [13]. Reasons for this are complex; for example, van Vliet et al. suggest that the nature of the

relationship between people with MS and neurologists may be important, as well as the association between palliative care and end of life, which speaks to the uncertainty inherent in MS. This clinical uncertainty is due to the unpredictable trajectory of MS, which includes periods of stability punctuated by crisis, and makes it challenging to identify people with MS who are approaching a point where mental capacity is compromised and/or where life may be limited [16–20]. Moreover, health professionals often fail to initiate these discussions, possibly due to their reluctance to discuss disease progression and death and their difficulty managing their own emotions during difficult conversations [21, 22]. Whereas ACP is well-researched in other patient groups [23–25], it has not been widely examined in relation to people with MS.

Additionally, while Lin et al. [26] developed a conceptual model of ACP in cancer patients, no theoretical understanding of the contexts or circumstances in which ACP is relevant to people with MS is yet available. Two questions warrant answers when examining ACP for people with MS: (i) "in what circumstances, with whom, how, and why do people with MS (and their families) engage in ACP?" and (ii) "what works for whom, how, and why, during ACP discussions?" This realist synthesis therefore aims to: identify core mechanisms generating engagement in and completion of ACP by people with MS (and their families); identify contextual factors that trigger (or inhibit) core mechanisms, and contribute to the theoretical understanding of the process of ACP in people with MS.

## Methods

The United Kingdom Medical Research Council's (MRC) guidance on development and evaluation of complex interventions [27] and the Methods of Researching End-of-life Care (MOR-ECare) statement [28] stress that new health care-related interventions are likely to be most effective when they are underpinned by a conceptual framework and a theoretical understanding of the key processes involved in delivering interventions and the contexts in which they are required to operate. This realist review specifically addresses the requirement for theory and conceptual framework development and was developed in April 2019 and published in PROS-PERO (registration number: CRD42019142294 https://tinyurl.com/ttdt963). We used a process adapted from the RAMESES publication standards [29] and adhered to Pawson's realist methodology [30]. The review drew upon the Integrated Behaviour Model (IBM) to guide theory development. Realist reviews are a theory-driven systematic approach that are particularly suited to helping understand causation; they aim to investigate what works (or fails to work) for whom, in what circumstances, and how, by identifying processes (mechanisms) that lead to desired outcomes in particular contexts [30, 31]. Furthermore, they examine how mechanisms or 'underlying causal forces or powers' are triggered in particular contexts and lead to outcomes [29]. This specifically relies on using *'context-mechanism-output'* configurations (CMOs); these represent testable hypotheses that explain the ways in which the context is able to trigger mechanisms and lead to a variety of outcomes [31] (Table 1 for glossary of terms).

The phases involved in conducting this realist review involved: (i) determining the scope of the review; (ii) formulating or articulating the key theory (iii) searching for key studies; (iv) selecting the studies and appraising their relevance; (v) extracting, analysing and then synthesising the data.

### Scope of the review

We clarified the scope of the review by engaging with key informants (clinical experts in neurology, palliative care and rehabilitation from different professional groups including nursing, medicine, physiotherapy and occupational therapy) and members of our study Patient and Public Involvement (PPI) group (people living with MS and their families). This process is

**Table 1. Glossary of realist terms used in the review.**

| Term | Explanation |
|---|---|
| Context | Pre-existing structures, settings, environments, circumstances or conditions that shape whether certain behavioural and emotional responses (for example mechanisms) are subsequently triggered. |
| Context-mechanism-outcome configurations (CMOs) | Describe the causal relationships between contexts, mechanisms and outcomes, that is, how certain outcomes are realised through mechanisms that are triggered in certain circumstances and contexts. |
| Mechanisms | The behaviour or emotional response that is triggered in certain contexts. The mechanism is context-specific and is usually hidden. |
| Outcomes | The final impact of mechanisms that are triggered in certain contexts. |

Adapted from Mitchell et al. [71] and Papoutsi et al. [92].

recommended for realist syntheses as it can help to shape the direction of the review by articulating themes and ideas relevant to ACP in MS [29, 32, 33]. During informal interviews, we asked key informants to identify salient contextual factors that encourage or discourage ACP discussions that included, but were not limited to, age of onset, symptoms, and salient concepts such as uncertainty or transitions. Interviews were not recorded, however, notes were taken during these meetings which helped to form the basis of our initial search strategy. PPI members identified the desire to discuss ACP with their health care provider but felt discouraged due to lack of information about their disease trajectory as well as time pressures at medical appointments. Health care providers identified uncertainty in the disease trajectory, an unwillingness to discuss death and dying with MS patients, and a presumption that patients did not want to discuss ACP or that ACP was not important to MS patients. Both the health care providers and PPI informants expressed confusion about how to initiate ACP discussions.

Based on the information received from key informants, we then conducted a scoping exercise to identify strategies that promote successful ACP in people with MS and other illness contexts that may share similar characteristics. We discovered a small body of literature on ACP within MS and. From these sources of evidence we gained a broad understanding of ACP to focus the review and develop key theories to underpin our main literature search.

## Articulation of key theories

A distinguishing feature of a realist review is the recognition of theoretical drivers for the intervention (ACP), which are scrutinized using evidence from the review. These drivers are the expectations and justifications for why an intervention might work. We developed our initial programme theory through 'creative thinking sessions' involving the key informant consultations, the scoping exercise results, and the use of substantive theory (the Integrated Behaviour Model (IBM), based on Azjen's Theory of Planned Behaviour [34, 35]). This substantive theory helped us to understand how people with MS and health professionals make decisions about ACP by explaining that behaviour changes are linked to underlying beliefs about the behaviour itself, associated outcomes, and an individual's ability to perform the behaviour. Motivation represents a key driver for behaviour change and is comprised of three elements: attitudes, perceived norms, and personal agency [36] which are influenced by context. We used the *'context-mechanism-outcome'* (CMO) heuristic to explore various configurations that led to the development six CMO hypotheses that reflected our initial programme theory (refer to Table 2).

**Table 2. 'Context-mechanism-outcome' hypotheses.**

| | Context | Mechanism | Outcome |
|---|---|---|---|
| 1. | If people with MS experience losses | then they will accept that MS is life-limiting and will come to see themselves as a person with a life-limiting illness | and they will be more likely to engage in ACP |
| 2. | If people with MS have a trusting and empathic relationship with their healthcare provider | then they will feel empowered | |
| 3. | If people with MS feel they are a burden to family members | then they will look for ways to reduce their family member's future decisional conflict, | |
| 4. | If people with MS want to establish control over their future, | then they will come to understand ACP as a tool for autonomy | |
| 5. | If health care professionals have the communication skills to engage in open, frank, and timely discussions | then this would inspire the confidence to discuss death and dying | which would facilitate ACP engagement and completion. |
| 6. | If people with MS have witnessed 'bad deaths', then they will fear dying | and will perceive ACP as a way to prevent a 'bad death' | thus, will be more likely to engage in ACP. |

## Search methods

Literature searches were conducted from inception to 12 August 2019, in MEDLINE, PsychInfo, CINAHL, Scopus, Web of Science, and Embase databases, using terms presented in Table 3. In keeping with realist methodology, our literature search process was multi-pronged and iterative [29, 33].

Additional search strategies included citation tracking on MEDLINE to identify related papers and the citations within identified papers for relevance and additional reference chaining. We also searched grey literature on the following United Kingdom websites: Multiple Sclerosis (MS) Trust, Multiple Sclerosis (MS) Society, National Health Service (NHS), Google and OpenGrey.

## Screening methods and inclusion criteria

Articles were screened for eligibility by two researchers (LC and GE) and selected based on predetermined eligibility criteria (Table 4) for their relevance to theory building and our

**Table 3. Search strategy for MEDLINE, PsychInfo, CINAHL, Scopus, Web of Science, and Embase.**

| Term 1 | Term 2 |
|---|---|
| Advance care planning | Multiple sclerosis |
| Advance* care plan* | MS |
| Advance directives | Neuro-degenerative diseases |
| Advance* directive* | Disability |
| Living will | |
| Decision making | |
| Goal setting | |
| Future | |
| Decision support techniques | |
| Treatment planning | |
| End of life conversation* | |
| End of life decision making | |
| End of life planning | |
| Palliative care | |
| Palliative care conversation* | |
| Attitude to death | |
| Future | |

**Table 4. Inclusion and exclusion criteria.**

| Inclusion criteria | Exclusion criteria |
|---|---|
| 1. Source focused on advance care planning (including end of life conversations, end of life decision making, palliative care conversations) | 1. Source focused on completion of advance directives or DNAR orders or on shared decision-making models |
| 2. Source focused on the following patient groups: | 2. Source focused on the following patient groups: |
|  | • cancer patients |
| • Patients living with multiple sclerosis | • living with chronic non-neurological illness |
| • Patients living with a significant physical disability | • patients living with intellectual/learning disability |
| 3. Source types | 3. Source not written in English or French |
| • All study designs and other forms of academic and grey literature, including literature reviews, editorials, and guidelines related to advance care planning |  |
| 4. Source is written in English or French |  |

During the screening we applied the following criteria: *"Does this data source provide any evidence, discussion, or conceptual/theoretical perspectives that will enable us to test and refine our understanding of engagement in advance care planning for People with MS"*?

research question [29, 37]. We revisited our criteria throughout the review process and amended them based on iterative literature searches and preliminary data extraction.

## Data extraction

To promote consistency, we (LC, GE and JK) developed a bespoke data extraction form and extracted information related to the six CMO hypotheses and information about contexts, mechanisms and outputs to promote consistency in the data extracted. Two researchers (LC and GE) then independently extracted data from all selected papers.

## Appraisal of included literature

We supported Pawson and Tilley's (1997) decision to reject the 'hierarchy of evidence' [38] approach to quality appraisal and adopted a system to ensure rigour according to the following principles: (i) faithfulness or correspondence with the programme theories; (ii) trustworthiness or fidelity: the evidence reported was reliable; and; (iii) relevance or application of data to the research aims. The relevance of each document was scrutinized in detail in terms of its ability to contribute to our CMO's, as well as its credibility or the extent to which its data supported its findings and their clinical utility and applicability [29, 38]. Additionally, for the qualitative studies we used established appraisal techniques [39], an approach used in other realist reviews [40], that enabled us to examine to what extent studies presented 'thick' or 'thin' accounts of the intervention's components and their respective contexts and mechanisms.

## Data analysis and synthesis

The data analysis and synthesis processes were flexible, iterative, and creative. To maintain transparency, LC, GE and JK kept notes from a series of meetings during which they discussed each article and its contribution to the CMOs. Moving between theory and data we used retroduction to explore, compare, and explain observable patterns in the data, seeking their essential conditions, while looking for and exploring non-observable data not captured by our initial programme theories (CMOs). We used abductive reasoning for the non-observable data to create associations and to recontextualise the data, creating new plausible conclusions [29, 41]. For both processes, researchers (LC, GE, and JK) discussed potential explanations, new findings, and strategies to refine and revise our CMOs. We used flipcharts to outline each

CMO and its supporting data, while evaluating its relevance to our findings. Relevant key informants were apprised of our preliminary findings and provided feedback. We then mapped our findings to a schematic of the IBM to synthesise our theoretical understanding of ACP. We retained the notes, flip charts, and schematic as an audit trail of decisions made. The final synthesis is an interpretive yet robust collation of the supporting evidence we located for each of our CMOs. To promote transparency, the data are presented in the findings section as direct quotations from the supporting literature [31].

# Results

## Study selection

Our searches yielded 4,034 articles. After removing duplicates, two researchers (LC and GE) scanned titles and/or abstracts for relevance, excluding 3,897 articles. They independently read full texts and excluded a further 64 articles resulting in 42 articles. Based on reviewer's comments, GE and LC scrutinised the included literature and excluded a further nine articles on neurological diseases other than MS, which did not add new data to our CMOs. This resulted in 33 final articles for inclusion in the review. The data screening processes is depicted in Fig 1.

## Study characteristics

Of the 33 articles, 17 focused on MS exclusively [20, 42–57] seven on MS in combination with chronic neurologic or non-neurologic illness [17, 58–63]; one on MND [64] seven on neurological illness (non-specified) [18, 19, 65–69], and one on disability [70]. Nineteen were qualitative or quantitative studies [20, 42–44, 46, 47, 49, 50, 54–56, 59, 61–64, 67, 68, 70]; one was a mixed methods study [19]; and 13 were case studies, literature reviews or discussion/opinion articles [17, 18, 45, 48, 51–53, 57, 58, 60, 65, 66, 69] of which seven were opinion or expert panel-based articles [17, 18, 48, 51, 53, 58, 66] three were case-studies [45, 57, 65] and three were literature reviews (narrative/systematic) [52, 60, 69] [refer to S1 File for study characteristics].

The source articles reflect an international scope; two had an international authorship [51, 60], seven from USA [43, 44, 46, 48, 66, 68, 70], seven from UK [17, 18, 52, 56–58, 65] seven from Canada [20, 49, 50, 55, 59, 62, 67], four from Germany [16, 47, 54, 63], two from the Netherlands [61, 69], one from Australia [64], Italy [53], Peru [19] and Turkey [45].

## Substantially supported CMO hypotheses

### CMO 1: Cumulative losses lead to acceptance of MS as a progressive condition and the creation of a new self-identity where ACP is relevant

For many people with MS this was identified as an important mechanism in both the pre-engagement and engagement phases associated with ACP. By experiencing clinically significant and person-centred losses, including physical functions, roles, paid employment and, in some cases losing touch with friends, people with MS underwent a transition. This markedly disrupted their biography and concept of self to increasingly include MS as more part of their identity. Consequently, they began to see their futures as progressively uncertain and ACP as more relevant to them. This new awareness led to an increased willingness to engage in ACP. This CMO was supported by ten studies; [20, 46–49, 53, 56, 62, 64, 67]. Lowden et al. [20] explored the process of "coming to a redefined self" (p.e17) in nine participants with relapsing-remitting MS. They discovered coming to accept themselves as a person with MS involved "one's sense of self-being called into question" (p.e21) associated with multiple losses, particularly previous roles. This redefinition of self was integral to decision-making since participants were not prepared to consider making treatment decisions until this transition occurred [20].

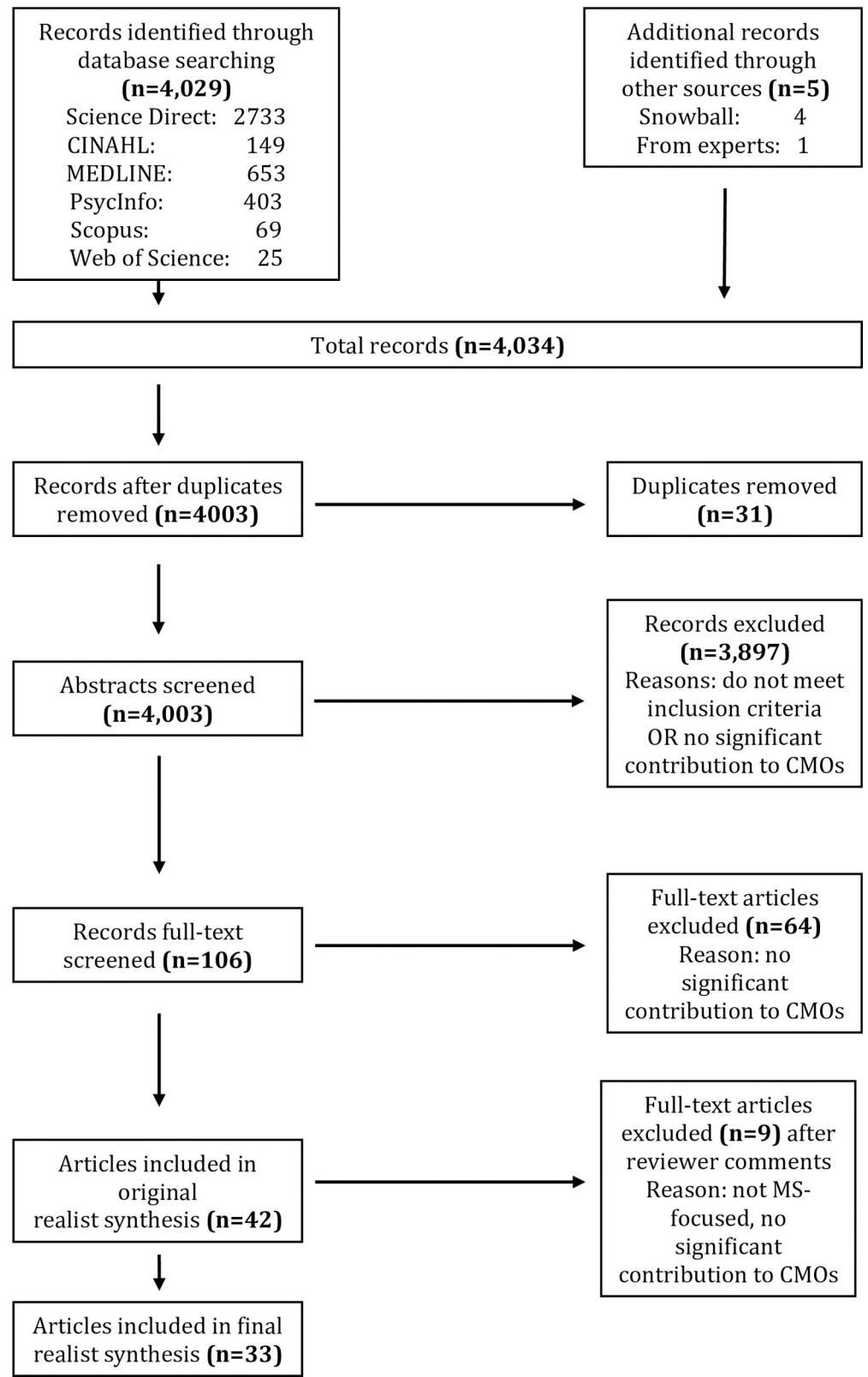

**Fig 1. Modified PRISMA flow chart of literature search.**

Similarly, Paterson et al. [62] observed that decisions about the future were emotionally charged and "implied submitting to the progression of the disease rather than accommodating a temporary setback" (p. 67), a view shared by Lustig et al. [46] that adjustment to loss was vital in people with MS. Murray et al. [64] go further to suggest "ACP was easier for patients and caregivers who accepted encroaching death" (p.473) but was also present at other stages of the disease, albeit to a lesser extent.

Acceptance as the mechanism that underpins optimal timing for ACP discussions can be challenging due, in part, to MS's uncertain trajectory and people with MS often being overwhelmed by the complexities of treatment initiation, balancing treatment side-effects and the need to learn and re-learn how to live with a progressive disease. Despite this, two articles recommended initiating discussions about ACP as early as possible, even at the point of diagnosis [60, 65], citing as justification future impaired communication or cognitive abilities. However, the most common recommendation to initiate discussions was after people with MS experienced key triggers indicating clinically significant disease progression [20, 49, 53, 57, 58]. Lowden suggests the rationale for this is that: "Participants felt ill-equipped to make treatment decisions early in the disease trajectory (p.e19). Those participants who had not yet reached the point of considering MS as part of their identity were therefore unwilling to receive information, education or decision support (p. e22)" [20]. Similarly, Leclerc-Loiselle and Legault described timing in the context of loss and acceptance: "Professionals all considered the adaptation period of people with MS to their new functional reality following an MS crisis was a pivotal moment to introduce a palliative approach. The moment right after an MS exacerbation was a turning point because there has usually been a reflexive process as a result of grieving the loss of previous functional abilities" [49] (p.267). This reflexive process of acceptance represents the mechanism that underpins potential optimal timing of ACP engagement in discussions in people with MS. Though none of the included studies discussed a definitive time frame, they did suggest that acceptance is a gradual process [20, 62].

## CMO 2: Trusting and empathic relationships enable empowerment

Building trust centred on active and reflective listening and validating patient's concerns and fears are foundational to engagement in ACP. Evidence from 10 studies substantiated this CMO hypothesis [20, 42, 44, 49, 50, 52, 55, 59, 63, 70]. A relationship grounded on trust and empathy was essential when engaging in ACP discussions. This provided a safe space empowering people with MS to share fears and hopes for the future. For example, Thorne et al. [55] noted "although actual communication practices were important, the orientation of the health care providers to the person with MS was even more fundamental" (p.11). This orientation "was centred on respect and trust and involved validation of the patient's experience" [p.17]. Some people with MS may approach the health professional-patient relationship with suspicion, based on the inherent complexity and length of the MS diagnostic process, during which they may have spent years feeling undermined and unheard by their health provider [50, 71].

Driedger et al. go further, emphasising the importance of trust and empathy in the relationship between people with MS and their neurologist, remarking "the crucial aspect . . . was not solely what was being said, but also how one was saying it" (p.10) [50]. Additionally, "empathy, respect, and a willingness to acknowledge the perspectives of people with MS" [50] were critical to ACP's success. Likewise, Col et al. [44] reported that "physicians can encourage or obstruct patient involvement. Patients are more active when interacting with physicians who engage in partnership building and supportive talk that legitimises the patient's perspective and creates expectations and opportunities for the patient to discuss their needs and concerns" (p.266). Buecken et al. [42] noted that "physician empathy plays a key role" (p.323) in meeting

end of life care-related discussions and is paramount in delivering high-quality care. Furthermore, the study observed trust in a patients' health professional was imperative in initiating shared decision when considering current and future treatment decisions. Mitchell [70] remarked: "a trusting relationship with healthcare providers was a prerequisite to ACP discussions" (p.130).

## CMO 3: Fear of being a burden to family motivates engagement in ACP

Six studies supported this CMO [20, 43, 65, 66, 69, 70]. The presence of family was important to both the people with MS and the health professional(s). People with MS relied on and welcomed the support and caregiving provided by family, but occasionally their need for caregiving only increased negative thoughts of being a burden [43, 64, 70]. Some people with MS considered ACP a way to alleviate feeling like a burden. Murray et al. observed, "many participants indicated that documented wishes would reduce the decisional burden and help caregivers avoid regret" (p. 475) [64]. Similarly, participants in Mitchell's study reported that future planning "constituted a personal responsibility to their family members" (p.128) and "expressed concerns about financial and emotional burdens their families may experience" (p.131) [70]. Caregivers in Murray's (2016) (p.475) [64] study also described ACP as a "catalyst for communication" to realise open family discussions and ease tension in family negotiations.

For health professionals, family were viewed as invaluable resources whose intimate knowledge of dependants had potential to help initiate timely ACP discussions. Similarly, Seeber stated "involving families in the discussion appears to improve acceptance of decisions for both patient and caregiver" (p.596) [69].

## CMO 4: ACP as a tool for enabling control and autonomy in decision-making

Ten studies supported this CMO hypothesis [17, 18, 48, 56, 57, 61, 65, 67, 70, 72], which is the basis underpinning the original development of concept of ACP. Kalb explained: "people living with MS struggle to maintain a feeling of control in the face of an unpredictable disease course and uncertain future. Effective planning and problem-solving [. . .] help people feel prepared—more in control—regardless of what the future holds" (p.532) [48]. Clarke et al. also acknowledged "some participants used advance planning as a way of extending the zone of personal autonomy and involvement in decision-making beyond the stage when their ability to make decisions or communicate their wishes would be lost" (p.6) [56]. Mitchell reported participants felt engaging in ACP "constituted a responsibility to themselves [. . .] and represented asserting agency over [their] care" (p.129) [70]. Family members in Murray et al.'s study similarly reported that ACP "gave (them) control of life" (p.474) [64]

## CMO 5: Skilled communication inspires confidence and facilitates ACP discussion

Confidence and communication skills were important mechanisms to facilitate ACP completion, reported in 16 source articles [18, 19, 42, 44, 45, 50, 51, 55, 58, 59, 60, 64, 67–70]. Strong evidence supported an inverse relationship between poor communication skills and the absence of ACP discussions. A participant with MS in Thorne et al.'s study stated: "communication with health care providers can shape the entire course of an illness" (p.10) [55]. Similarly, Driedger and colleagues reported, "how things are said are as important as what is being said" (p.10) [50]. Health professional distance and paternalism were perceived as barriers to engaging in clinical decision-making processes [59]. These attitudes, enacted through

behaviours such as using simplistic explanations or deliberately obfuscating with highly technical scientific jargon, were perceived as being counterproductive. As Driedger et al. discovered, "a core problem was that neurologists just generally lack a capacity for good communication and compassionate care" (p.10) [50] possibly a result of health professionals lacking clarity on the situation themselves. Thorne et al. identified withholding information was highly frustrating for people with MS: "I just get the feeling that it's a very paternalistic sort of attitude [. . .] 'We don't need to tell her that" (person with MS) (p.12) [55].

Beneficial communication included strategies that included legitimising and confirming the person with MS's experience, considering MS in the context of that person's life, and assisting them to find the language to describe their illness situation [55]. Col et al. observed that "physicians can encourage or obstruct patient involvement [in future care planning]" (p.266) [44]. Additionally, Oliver et al. observed: "families do appreciate honesty and awareness of deterioration" (p.36) [60].

However, we identified only limited agreement on which health professional was best placed to have ACP discussions with people with MS. Seeber [69] stated that the neurologist is "best-poised' due to their detailed knowledge of neurologic diseases and possible complications" (p.595). Surprisingly, our findings indicate that knowledge was less valued than the ability to establish and maintain what people with MS considered to be a trusting relationship. Our recommendation is therefore that the person best positioned to have discussions with people with MS would be the one with whom they believe to have the strongest and most trusting relationship, a situation that is open to change.

## CMO 6: Previous experiences of witnessing death facilitates or hinders engagement in ACP

Seven studies corroborated this CMO [20, 43, 47, 49, 56, 65, 71]. ACP was reported to mitigate the fear of experiencing a distressing or 'bad' death [43, 65] and may be a motivating factor for some people with MS. For example, in Chen and Habermann's study, participants shared views that having witnessed a 'bad death' or having experienced a life-threatening illness in a relative acted as a "close call" (p.4) that triggered their own fear of dying and the need to address it through engaging in ACP. Former health professionals were more likely to have witnessed bad deaths, resulting in the acknowledgement that ACP may represent a way to address their concerns [20, 43]. Lowden et al. stated, "previous experiences with illness and treatment influenced participants' decision making" (p.19) [20].

In addition to these core mechanisms, we identified two related components: the fear of discussing death and dying and the concept of hope in MS. The fear of discussing death and dying was present in four articles [20, 47, 65, 70] and may be considered to be wide-spread in post-modern Western culture [65, 73]. Related to this, the reviewed literature identified that health professionals were reluctant to initiate ACP discussions because they feared discussions would lead to the loss of hope in the person with MS [19, 20, 49, 64]. Correspondingly, some people with MS were reluctant to engage in ACP because they held out expectations of a cure for their MS [56]. As Clarke et al. observed: "Decisions about future care were deferred to a later date, sometimes in the apparent hope that eventualities that they could plan for may never arise" (p.7) [56].

## Discussion

We present the first realist review to develop and refine an initial theory to explain engagement in and completion of ACP in people with MS and to guide clinical practice by uncovering in what ways ACP works. Specifically, by testing our six CMOs, we identified mechanisms

underpinning the behavioural changes that drive engagement in, and completion of, ACP. They included: acceptance, experience, confidence, empowerment, fear (of being a burden; of death and dying) and the desire for autonomy. In the USA, Levi et al. [74] explored motivations that led people to engage in ACP. The authors explained that motivation to engage in ACP was contingent on four distinct domains: concern for self; concern for others; expectations about the impact of ACP and anecdotes, stories, and experiences. Their findings share similarities with this review; a number of the CMOs we developed and tested could be similarly categorised. For example, 'concern for others' aligns closely with our CMO about burden on family; 'concern for self' parallels the desire for autonomy and the fear of suffering a 'bad death'. The influence of prior experiences on ACP motivation echoes Levi et al.'s category of 'anecdotes, stories, and experiences', a finding similarly identified by Shaw et al. [75] in a study of ACP in long-term care facilities. They observed "both residents and family members indicated that prior experience with [. . .] the advance care planning process had helped in influencing current perspectives and readiness. As a result, both groups described having engaged and feeling ready to engage in the process" (p.745). However, neither Levi et al. [74] nor Shaw et al. [75] addressed the mechanisms underlying the motivations for engagement in ACP: they identified what the motivations were but did not identify why these mechanisms work or how to trigger them.

We discovered acceptance of self as a person with a life-limiting illness is the core mechanism underpinning engagement in ACP for people with MS. Acceptance facilitates people with MS to see the relevance of ACP and is a driver for subsequent behaviour change, which is the willingness to engage in ACP discussions. According to the IBM, relevance aligns with attitudes, which are a key facet of behaviour change [76]. Attitudes refer to the individual's perception of the behaviour; in this case, acceptance of self as a person with progressive illness changes their perception of ACP from irrelevant to relevant. Relevance is critical for ACP. Although Solari et al. [14] recently reported that as many as 89% of people with severe MS want to discuss ACP, preliminary findings from a study on ACP in MS indicate that while participants were generally positive about ACP, it was no more relevant to them than to anybody, regardless of their MS diagnosis. Concerning their MS specifically, participants were reluctant to engage in ACP because of the uncertainty of their disease trajectory [15]. The differences in interest in ACP may be due to levels of disease severity, as evidence suggests the notion of ACP often becomes relevant to people with MS when disease severity increases [43, 56].

Questions still remain about how and when to make ACP relevant to those living with, and in some instances dying from the complications associated with MS. Careful and sensitive encouragement of people with MS to accept their situation will help, but we as identified, an important challenge encountered with the uptake of ACP is the actual context in which it is enacted. We initially presumed that two related contextual elements—the fear of discussing death and dying and hoping for cure—would influence consideration of ACP. However, we underestimated their importance. Indeed, both elements created contexts that were largely inhospitable to ACP and represent fundamental issues with its uptake. The mechanisms we uncovered, while important, may not sufficiently overcome these issues, but may function instead as 'trust mechanisms' [32]. Trust mechanisms work to influence contexts rather than influencing outcomes directly; in this case, mechanisms enable contexts to be more hospitable to notion of ACP. Trust mechanisms alter the shape of traditional CMO configurations, changing from $C + M = O$ to $C + M \rightarrow C \rightarrow O$.

For clinical practice, work must be aimed at reshaping the context, rather than focusing strictly on outcomes associated with behaviour change or 'completed' ACPs. Each mechanism we identified can be leveraged to change the landscape for people with MS, to facilitate acceptance of ACP if that is their preference. For example, relationship building may open a trusting

platform on which more accurate and realistic prognostic information can be shared with people with MS; this is essential to help them make sense of their illness and reframe their self-image. Recent European Academy of Neurology (EAN) guidelines for palliative care in MS include early discussion of disease progression and future planning with people with severe MS [14]. Realistic hope is also important; our findings indicate health professionals do not discuss ACP for fear of distressing people with MS. However, false hope may undermine trust, whereas honesty, empathy, and truthfulness are valued by people with MS [50].

Additional to building trusting relationships, advanced communication skills are important for health professionals, many of whom are notably uncomfortable discussing death and dying [19, 67, 68], exacerbating an already difficult conversation. Communication skills training linked to patient-centred outcomes are recommended [77]. This must be accompanied by active listening [44, 55] and the use of accessible language [50]. Andreassen et al. [78] utilised discourse analysis to explore ACP discussions, using the concept of doctor and patient voices. They discovered that health professionals used a variety of 'voices': the 'doctor voice' was used to ask specific questions; the 'educator voice' to share information and help patients understand their illness and treatment, and the 'fellow human voice' to convey empathy. Simply by showing empathy through comments for example "I understand" or "that must be really tough", health professionals shared a 'fellow human voice' encouraging patients to discuss ACP.

Additionally, in an attempt to develop a measurement tool for ACP, several author groups mentioned "prerequisites" [79, 80]. These prerequisites, such as personality factors, cognitive style, coping style, role preferences, risk knowledge, numeracy, risk attribution and tolerance, should also be recognized by healthcare professionals as they are part of the patient's personal context and might influence the communication pathways to be engaged in this communication relationship.

Overall, this realist synthesis uses a theory-testing method to explore, below the surface, what helps or hinders ACP from taking place and to understand *how* contextual barriers and facilitators operate. This knowledge is critical to design evidence-informed interventions that have the greatest potential to influence underlying mechanisms that drive behaviour change in different contexts and lead to positive person and family-centred outcomes and experiences.

## Strengths and limitations

A strength of this approach is both its explanatory and theoretical nature through which to understand the complex mechanisms underlying pre-engagement and engagement of ACP in people with MS. We also actively incorporated key informants' and PPI views alongside published literature to refine the review focus to areas considered most pertinent to clinical practice and lived experiences of MS. Moreover, this style of synthesis shifts the focus from specific interventions and services to broader underlying mechanisms or principles. Additionally, our evidence included a diverse range of clinical conditions relevant to people living with MS. Thirteen of the studies were qualitative, a strength being that this permitted salient contexts, mechanisms and outcomes to be understood in detail, particularly where 'thick' accounts were evident. Whilst we were able to draw on a relatively wide international literature, we are aware that ACP is still a predominantly Western construct [81, 82]. Cultural homogeneity is therefore a limitation of this review. Caution must therefore be exercised when considering the findings of this review for countries where ACP has yet to be implemented.

Institutional factors are undeniably part of the context of ACP, as health care is embedded in discourses of efficiency [83], and the overriding dominance of 'cure-over-care' [84] in which palliative care, including ACP, can often be disregarded. Our findings do not extend

across multiple levels (micro, meso, macro), but are focused principally on the individual. Since we set out to study pre-engagement and engagement phases of ACP, rather than its implementation, we believe focus on the individual is justifiable, although we recognise a multi-level exploration and analysis could provide complementary evidence [81].

## Conclusions and recommendations

We have identified important mechanisms that help to explain how and why ACP works in people with MS. Additionally, based on our findings we suggest strategies to assist health professionals help people with MS to engage in a process where they are able to reframe, perhaps multiple times, their self-image, so that ACP becomes more relevant to them. This research is important, as it is evident that ACP is more than merely an administrative or intellectual task for health professionals to engage in with their patients. Instead, it is a deeply human encounter in which a person reflects on their illness, situation, on-going rehabilitation and management, and impending death and must negotiate a route through at times threatening possibilities.

Consequently, health professionals have a responsibility to train in communications skills [77] and specifically in engaging in difficult conversations [85]. Evidence suggests that people with MS prefer an active role in decision-making [86, 87], therefore health professionals must work towards empowerment through partnerships. Moreover, they must appreciate that offering patients accurate and honest information about the patient's prognosis is unlikely to destroy their hope and may paradoxically encourage realistic future planning. Importantly, this can only be fostered if authentic relationships are built with patients and their families; creating environments in which respect, trust, and empathy thrive is fundamental to shape a context that supports engagement in ACP discussions.

Future research must now take place that that informs not only the evaluation of effectiveness of ACP among people living with MS and their families, but also actively engages in the CMOs examined in this review to better understand how this complex intervention can successfully and acceptably be enacted. The 'gold-standard' randomised controlled trial design alone does not adequately address the 'context-specific drivers' behind implementation outcomes and their relationship to the underlying theory. We therefore suggest researchers consider using a realist approach alongside more conventional designs akin to a 'hybrid trial' design that includes multiple methods. Realist evaluation is increasingly applied in the examination of complex healthcare interventions, for example ACP, since it seeks to provide a more in-depth understanding of what works, for whom and in what circumstances [38, 88]. Hybrid trials not only focus on assessing clinical effectiveness, but explore the manner in which an intervention is implemented [89, 90], and it has been argued should also consider context [91].

**Disclaimer:** The views expressed in this publication are those of the author(s) and not necessarily those of the NHS, NIHR, the Department of Health or the MS Society.

## Supporting information

**S1 File. Study/Literature characteristics.**
(DOCX)

## Author Contributions

**Conceptualization:** Laura Cottrell, Guillaume Economos, Catherine Evans, Eli Silber, Rachel Burman, Richard Nicholas, Bobbie Farsides, Stephen Ashford, Jonathan Simon Koffman.

**Data curation:** Laura Cottrell, Guillaume Economos, Jonathan Simon Koffman.

**Formal analysis:** Laura Cottrell, Guillaume Economos, Jonathan Simon Koffman.

**Funding acquisition:** Catherine Evans, Eli Silber, Rachel Burman, Richard Nicholas, Bobbie Farsides, Stephen Ashford, Jonathan Simon Koffman.

**Investigation:** Laura Cottrell, Guillaume Economos, Jonathan Simon Koffman.

**Methodology:** Laura Cottrell, Guillaume Economos, Catherine Evans, Jonathan Simon Koffman.

**Project administration:** Jonathan Simon Koffman.

**Supervision:** Jonathan Simon Koffman.

**Validation:** Laura Cottrell, Guillaume Economos, Jonathan Simon Koffman.

**Visualization:** Laura Cottrell, Guillaume Economos, Jonathan Simon Koffman.

**Writing – original draft:** Laura Cottrell, Guillaume Economos, Jonathan Simon Koffman.

**Writing – review & editing:** Laura Cottrell, Guillaume Economos, Catherine Evans, Eli Silber, Rachel Burman, Richard Nicholas, Bobbie Farsides, Stephen Ashford, Jonathan Simon Koffman.

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
