## [Decision Letter · Decision Letter 0]

5 Jun 2020

PONE-D-20-09582

A realist review of advance care planning

for people with multiple sclerosis and their families

PLOS ONE

Dear Dr. Koffman,

Thank you for submitting your manuscript to PLOS ONE. After careful consideration, we feel that it has merit but does not fully meet PLOS ONE’s publication criteria as it currently stands. Therefore, we invite you to submit a revised version of the manuscript that addresses the points raised during the review process.

Please answer these questions: 

Methods

The scope of the review section needs re-considering – if you have engaged with stakeholders evidence is needed of how and why and what the results are. You also mention a rapid literature review which is a type of review in itself – please reconsider this and what information is placed here. 

Please reference your searching process so it can be checked 

An audit trail is needed to show how you have progressed from data to the final synthesis. It needs to illustrate all stages – of Particular importance is integration because you have mixed methods data? Also you talk about theory being considered Pg7 how did it contribute to analysis?

Please check and identify what is the quality appraisal used for?

Results

Should the results be split by condition types? as mentioned on the end of page 10 and start of 11?

You have 26 experimental studies – but the results look like themes with no experimental data? You have four reviews not sure if it is clear how this evidence is integrated into the results? 

As a style a lot of it is about presented quotes – can a justification for your style of presentation and analysis of results be made up front 

I don’t see where critical appraisal findings are presented or why they are used?

We look forward to receiving your revised manuscript.

Kind regards,

Andrew Soundy

Academic Editor

PLOS ONE

Journal Requirements:

'This review is part of a larger study that was generously funded by the MS Society [Grant

code 93]. CE is funded by HEE/NIHR Senior Clinical Lectureship (ICA-SCL-2015-01-001'

'N/A'

Additional Editor Comments (if provided):

Reviewers' comments:

Reviewer's Responses to Questions

**Comments to the Author**

1. Is the manuscript technically sound, and do the data support the conclusions?

Reviewer #1: Yes

2. Has the statistical analysis been performed appropriately and rigorously? 

Reviewer #1: N/A

3. Have the authors made all data underlying the findings in their manuscript fully available?

Reviewer #1: Yes

4. Is the manuscript presented in an intelligible fashion and written in standard English?

Reviewer #1: Yes

5. Review Comments to the Author

Reviewer #1: The authors present a realist review on ACP in people with MS (pwMS). The paper is well written and presents a comprehensive review of the relevant literature for MS as well as other neurological diseases and physical disability. The fact that these conditions are quite heterogeneous and pwMS are hardly comparable with e.g. people with PD, HD, ALS or even dementia or COPD limits the meaningfulness of the analyses. Even though MS might be considered a “life limiting” disease, it surely is not for all people with MS as a considerable number will not experience a limited life expectancy and most will have at least decades with the disease and die at retirement age. I can see that the authors are unable to rerun the whole analysis focussing only on the few study with pwMS, but they should give very clear justification for choosing this broad view and clearly state this as a limitation.

Apart from this major limitations, there are some further aspects that should be addressed within a possible revision.

(1) In the introduction, it is stated that few pwMS engage in discussions about their future, which needs to be proven. The cited references from other neurological diseases are not particularly helpful. The recent paper by Köpke et al. (Eur J Neurol. 2019;26(1):41-50) at least gives some indication that they do. Also the work by Solari and Giovannetti and others on progression to SPMS and palliative care needs could be interesting here. Finally the soon to be published EAN-guideline on palliative care in MS could provide some more specific and important background information. Also the background on information provision and on health care professionals’ abilities to provide information would profit from MS-specific literature e.g. from the groups of Heesen and/or Solari. Finally, as cancer more and more becomes a chronic condition, it should be made clearer why this would be expected to clearly differ from MS. Especially as in the discussion (p.21) the authors refer to people with cancer when it comes to “trust”.

(2) Under “study characteristics” it seems that the first citation should be [15] and not [13]. Also, I cannot see that there are 26 “experimental” studies. At least, I would expect a definition of “experimental” here.

(3) The results section is surely a strength of the paper and the description of the CMOs is mostly transparent and clear. At the end of CMO 5, it is stated that the person that will have ACP discussions with the pwMS should be a trustworthy person, but knowledge and skills seem less or even not important. This does not seem to fit the concept of ACP.

(4) In the second paragraph of the discussion, the authors refer to pwMS’ “subsequent behaviour change” and I am unsure what this refers to. In the same paragraph the final aspects claiming that only few people see the relevance of ACP can surely not be made with a paper more than 10 years old, considering the dynamic development concerning ACP. I like the CMO formula on p.21, but think that figure 2 should be deleted as it does not contain any further information.

(5) As stated above, under “Strengths and limitations”, the fact that “a diverse range of clinical conditions” are included should be stated as limitation as it can surely be doubted that these are “relevant to people living with MS”.

(6) In the conclusion section, I agree that pwMS should be accurately and honestly informed and there is good evidence that this does not lead to harms although frequently feared by health care professionals. Here again the work by Heesen and/or Solari seems relevant including the recently updated Cochrane review on “Information provision…”. In this context, I wonder why the concepts of “shared decision making” and “evidence-based patient information” are not addressed. Also, the point that RCTs are principally not suitable is not convincing, considering the rich discussion on the evaluation of complex intervention e.g. within the MRC framework(s) cited in the introduction.

(7) Finally, I wonder why existing ACP frameworks and interventions such as “Respecting Choices” are not discussed for pwMS.

6. PLOS authors have the option to publish the peer review history of their article (what does this mean?). If published, this will include your full peer review and any attached files.

Reviewer #1: No

---

## [Author Response · Author response to Decision Letter 0]

26 Aug 2020

EDITOR – RESPONSE TO EDITOR

1. The scope of the review section needs re-considering – if you have engaged with stakeholders evidence is needed of how and why and what the results are.

Based on realist evaluation standards (Wong et al., 2016), we met with key informants (or stakeholders) to shape and re-shape CMOs, ensure their relevancy and identify similarities with other conditions that might be used to broaden our understanding of the research question, and to clarify that our preliminary findings were appropriate to both clinical experts and people living with MS. On page 7 of the revised manuscript we have provided evidence of why and how we engaged with key informants of the revised manuscript and listed the results from the informal interviews.

2. You also mention a rapid literature review which is a type of review in itself – Please reconsider this and what information is placed here.

This is an excellent point; thank you. We have changed the terminology from ‘rapid literature review’ to ‘scoping exercise’ to reflect more closely the type of search we completed at the outset of this review process (please refer to page 8 of the revised manuscript)

3. Please reference your searching process so it can be checked.

Our search process is explained in more detail on page 9 of the revised manuscript and we have included an example Medline search strategy as Appendix A, S1. It is important to note however, that unlike a typical literature search for a systematic review, literature searches for realist syntheses are multi-pronged and iterative, conducted in response to emerging data (Booth et al., 2019; Wong et al., 2016). There is not one definitive search, but a series of searches undertaken throughout the data collection and preliminary analysis processes. 

4. An audit trail is needed to show how you have progressed from data to the final synthesis. It needs to illustrate all stages – of particular importance is integration because you have mixed methods data? 

With respect to an audit trail, we have updated the manuscript on page 11 of the revised manuscript (in the data analysis and synthesis section) to include a description of our process. 

5. Also you talk about theory being considered pg 7 how did it contribute to analysis?

The theory being considered (the Integrated Behaviour Model [IBM]) was used to aid in the development of our context-mechanism-output configurations (CMOs). Realist syntheses are driven by theory, which then helps to clarify or uncover mechanisms that are not context-bound and are, therefore, transferable between settings or contexts (Wong et al., 2016). The IBM contributed obliquely to the analysis through its use in the development of our CMOs. This is presented on page 8 of the revised manuscript.

6. Please check and identify what is the quality appraisal used for?

As stated in the Appraisal of Included Literature section (page 10 of the revised manuscript), we supported Pawson and Tilley’s (1997) realist tenet to reject the concept of ‘hierarchical evidence’ and instead to rely on an appraisal of the relevance of each article’s findings to our research question and to our developing theory (Wong et al., 2016).

7. Should the results be split by condition types? As mentioned on the end of page 10 and start of 11?

After further consideration and reviewer comments, we have eliminated nine articles on conditions other than MS, as the reviewer suggested these may be too heterogeneous or potentially too diffuse to contribute meaningfully to our analysis. We have now narrowed down the sample of papers to include those principally focused on MS. Therefore, our results are now focused more clearly on MS alone. We did, however, retain one article on motor neurone disease (Murray et al., 2016) and one on physical disability (Mitchell, 2017) (please refer to our rationale below in our reply to the first comment from Reviewer 1).

8. You have 26 experimental studies – but the results look like themes with no experimental data? You have four reviews not sure if it is clear how this evidence is integrated into the results.

We have changed the terminology (please refer to page 12 of the revised manuscript and in the supporting documents) with which we refer to our included articles, as “experimental” was indeed confusing and did not accurately describe the literature. We hope that the types of included articles are now more clearly defined with the following terms: “qualitative, quantitative, case, or mixed-methods study”, “discussion/opinion” article, and “literature review”.

Our “themes” are in fact CMOs: realist reviews do not report experimental data as such; instead, in a realist synthesis pre-defined CMOs are evaluated for their relevance to the data. CMOs differ from “themes” presented in qualitative literature, and are instead a tool based on an “If/then” statement in the following configuration: “If [context], then [mechanism], leading to or resulting in [outcome]” (Wong et al., 2016).

9. As a style a lot of it is about presented quotes – Can a justification for your style of presentation and analysis of results be made up front?

The presented quotes (in the results section) are data; for transparency, we chose to present them as examples of direct CMO-supporting data from the literature. This is an accepted practice in realist syntheses (Wong et al., 2016; Flynn et al., 2018). In the revised manuscript, we have addressed this on pages 10-11 in the data analysis and synthesis section.

10. I don’t see where critical appraisal findings are presented or why they are used?

In realist methodology, quality appraisal is based on relevance—rather than on the ‘hierarchy of evidence’ associated with more traditional systematic reviews (Pawson & Tilley, 1997; Wong et al., 2016). As stated in the quality appraisal section (now changed to the ‘Appraisal of Included Literature’ section; page 10 of the revised manuscript), we followed Pawson and Tilley’s (1997) advice and included articles if they contributed meaningfully to our CMOs.

11. In your data availability statement, you have not specified where the minimal data set underlying the results described in your manuscript can be found. Plos defines a study’s minimal data set as the underlying data used to reach the conclusions drawn in the manuscript and any additional data required to replicate the reported study findings in their entirety. All Plos journals require that the minimal data set be made fully available. for more information about our data policy, please see http://journals.plos.org/plosone/s/data-availability

The minimal data set used to reach the conclusion is provided as a table included in the ‘Appendix A & B, Supplementary File 1’.

REVIEWER #1 - RESPONSE TO REVIEWER #1

1. The authors present a realist review on ACP in people with MS (PwMS). The paper is well written and presents a comprehensive review of the relevant literature for MS as well as other neurological diseases and physical disability. The fact that these conditions are quite heterogenous and PwMS are hardly comparable with e.g. people with PD, HD, ALS or even dementia or COPD limits the meaningfulness of the analyses. Even though MS might be considered a “life limiting” disease, it surely is not for all people with MS as a considerable number will not experience a limited life expectancy and most will have at least decades with the disease and die at retirement age. I can see that the authors are unable to rerun the whole analysis focussing only on the few study with PwMS, but they should give very clear justification for choosing this broad view and clearly state this as a limitation.

Thank you; this is a very fair comment. We chose a realist review because it is a methodology that aims to establish causality between an outcome and events by uncovering the underlying mechanism leading from the events/context to the outcome (Wong et al., 2016). We chose this method because our ultimate aim is to develop an intervention designed to trigger this mechanism; our purpose in this synthesis was to uncover the mechanism(s).

After first focusing our search on MS, we wanted to deepen our exploration of our research question by focusing on those characteristics of MS that could contribute to the situation and constituted the “core” of our context. We were curious to know (a) what these were in people with MS, and (b) if they were unique to people with MS. To do this, we summarised our first results and consulted with our key informants who identified “uncertainty in the disease trajectory”, “uncertainty regarding future autonomy” and “uncertainty regarding acute relapses and their potential to recover” as important characteristics involved in engaging in ACP discussions for people with MS, in addition to “fear of discussing death and dying” with people with MS or “not knowing how to initiate a discussion about ACP”. Then, in collaboration with key informants, we identified that many neurological conditions and COPD share these characteristics with MS and decided to include these conditions, but only when considering the afore-mentioned characteristics. We also included physical disability, which shared many similar characteristics with and for many is a component of living with MS.

This process helped to broaden our understanding of the mechanisms that can occur in MS by relying on similarities with other conditions and is also a tenet of realist methodology (Wong et al., 2016). 

However, following your comments, we re-read all included articles on other conditions and reassessed their usefulness to our review. After careful consideration, we made the decision to exclude this literature (9 source articles) from our final CMO analysis, with two exceptions. We decided to retain the study conducted by Murray et al., 2016 (motor neurone disease) because it is relevant to three CMOs and, importantly because it explores in-depth the concept of acceptance, which is salient to the context of MS. We also retained the study conducted by Mitchell et al. 2017 because many people with MS also live with physical disability.

Apart from this major limitation, there are some further aspects that should be addressed within a possible revision.

2. In the introduction, it is stated that few pwms engage in discussions about their future, which needs to be proven. The cited references from other neurological diseases are not particularly helpful. The recent paper by Kopke et al. (eur j neurol. 2019; 26(1):41-50) at least gives some indication that they do. Also the work by Solari and Giovannetti and others on progression to spms and palliative care needs could be interesting here. Finally the soon to be published EAN-guideline on palliative care in MS could provide some more specific and important background information. Also the background on information provision and on health care professionals’ abilities to provide information would profit from MS-specific literature e.g. from the groups of Heesen and/or Solari. Finally, as cancer more and more becomes a chronic condition, it should be made clearer why this would be expected to clearly differ from MS. especially as in the discussion (p. 21) the authors refer to people with cancer when it comes to “trust”

We have revised the manuscript (on pages 4-5 of the revised manuscript) by removing the references from other neurological diseases and by incorporating more recent evidence to support our discussion about PwMS’ views on discussing ACP, including the Kopke et al. (2019) paper, a 2016 study on palliative care in neurology (including MS), and preliminary findings from an in-progress study funded by the UK Multiple Sclerosis Society.

After careful consideration we have eliminated the reference to cancer patients on page 21 of the first iteration of the manuscript to focus more explicitly on patients with MS. 

3. Under “study characteristics” it seems that the first citation should be [15] and not [13]. Also, I cannot see that there are 26 “experimental” studies. At least, I would expect a definition of “experimental” here. 

We adjusted this section after excluding the non-MS literature and have revised the citation numbers accordingly.

We have changed our terminology throughout the entire revised manuscript to reflect a more accurate description of the included literature; “experimental” was indeed a confusing term. We hope that the types of included articles are now more clearly defined with the following terms: “qualitative, quantitative, case, or mixed-methods study”, “Discussion/opinion” article, and “literature review”. 

4. The results section is surely a strength of the paper and the description of the CMOs is mostly transparent and clear. At the end of CMO5, it is stated that the person that will have ACP discussions with the PwMS should be a trustworthy person, but knowledge and skills seem less or even not important. This does not seem to fit the concept of ACP.

Thank for this comment. We agree. This does not seem to fit the concept of ACP and we too were surprised by what we found in the data. Our results suggest that people with MS would be more likely to engage in ACP discussions with someone trusted than with someone considered as knowledgeable. In the revised manuscript, we have altered the text on page 18 of the revised manuscript to reflect this.

5. In the second paragraph of the discussion, the authors refer to PwMS’ “subsequent behaviour change” and I am unsure what this refers to.

We have defined the ‘subsequent behaviour change’ as “willingness to engage in ACP discussions” in the revised manuscript on page 21 to clarify this; thank you very much for bringing this to our attention.

6. As stated above, under “strengths and limitations”, the fact that “a diverse range of clinical conditions” are included should be stated as a limitation as it can surely be doubted that these are “relevant to people living with MS”.

Agreed; we have excluded the articles (with two exceptions) that did not pertain exclusively to MS.

7. In the conclusion section, I agree that PwMS should be accurately and honestly informed and there is good evidence that this does not lead to harms although frequently feared by health care professionals. Here again the work by Heesen and/or Solari seems relevant including the recently updated Cochrane review on “information provision …”. 

We have revised the manuscript on page 25 to include the work from Heesen et al. and Solari et al. on the importance of active decision-making and partnerships for PwMS

We have also revised the manuscript on page 23 to include work from Heesen et al and Sudore et al. on patient “prerequisites” that may contribute to the communication context. Thank you for bringing our attention to this important work

8. In this context, I wonder why the concepts of “shared decision making” and “evidence-based patient information” are not addressed.

Again, you bring up an interesting point, and one that we explored rigorously throughout our review process. We did not include the shared-decision making literature for two main reasons. First, our research question was based on decision making in the context of advanced care planning. The process of completing an advance care plan requires that the patient have the ability to face the future and make decisions based on the wishes of a future unknown self. Contrary to this, shared decision making usually focuses on treatment decisions, which are immediate rather than future decisions. In addition, shared decision making differs contextually from ACP, as it does not intervene at the same point in the disease trajectory. Often patients making treatment decisions are newly diagnosed, whereas ACP discussions tend to involve patients who have been living with MS for years, if not decades. Making treatment decisions for immediate care is a significantly different experience than planning for future care and, possibly, impending death. Although at many points we considered including shared decision-making literature, we decided that for these reasons, it detracted contextually too far from our research question to add salient evidence to our CMOs.

9. Also the point that RCTs are principally not suitable is not convincing, considering the rich discussion on the evaluation of complex intervention e.g. within the MRC framework(s) cited in the introduction.

In the conclusion (on pages 25-26 of the revised manuscript) we state that RCTs do not adequately address the context-specific drivers. We do not mean to imply that RCTs are not suitable, but that complementary methodologies exist for exploring complex interventions, such as ACP. We have revised the paragraph (pages 26 of the revised manuscript) to clarify this point.

10. Finally, I wonder why existing ACP frameworks and other interventions such as “Respecting Choices” are not discussed for PwMS.

Although the “Respecting Choices” intervention may have applicability for PwMS, for the purposes of this review whilst we broadly explored grey literature, we confined the evidence to support our CMOs to the MS and neurology literature. In this way we hoped to discover contextual factors and uncover mechanisms unique to this patient population.

References 

Booth, A., Briscoe, S., & Wright, J.M. (2019). The ‘realist search’: A systematic scoping review of

current practice and reporting. Research Synthesis Methods. https://doi.org/10.1002/jrsm.1386. 

Flynn, R., Newton, A.S., Rotter, T., Hartfield, D., Walton, S., Fiander, M., & Scott, S.D. (2018).

The sustainability of Lean in pediatric healthcare: a realist review. Systematic Reviews,

7, 137.

Pawson, R., & Tilley, N. (1997). An introduction to scientific realist evaluation. In E. Chelimsky &

W.R. Shadish. (Eds.). Evaluation for the 21st Century: A Handbook. (p. 405-418). 

Sage Publications Inc. https://doi.org/10.4135/9781483348896.n29

Wong, G., Westhorp, G., Manzano, A., Greenhalgh, J., Jagosh, J., & Greenhalgh, T. (2016).

RAMESES II reporting standards for realist evaluations. BMC Medicine, 14, 96.

---

## [Decision Letter · Decision Letter 1]

1 Oct 2020

PONE-D-20-09582R1

A realist review of advance care planning for people with multiple sclerosis and their families

PLOS ONE

Dear Dr. Koffman,

Thank you for submitting your manuscript to PLOS ONE. After careful consideration, we feel that it has merit but does not fully meet PLOS ONE’s publication criteria as it currently stands. Therefore, we invite you to submit a revised version of the manuscript that addresses the points raised during the review process.

Please attend the reviewers very minor concerns. 

We look forward to receiving your revised manuscript.

Kind regards,

Andrew Soundy

Academic Editor

PLOS ONE

Reviewers' comments:

Reviewer's Responses to Questions

**Comments to the Author**

1. If the authors have adequately addressed your comments raised in a previous round of review and you feel that this manuscript is now acceptable for publication, you may indicate that here to bypass the “Comments to the Author” section, enter your conflict of interest statement in the “Confidential to Editor” section, and submit your "Accept" recommendation.

Reviewer #1: (No Response)

2. Is the manuscript technically sound, and do the data support the conclusions?

Reviewer #1: Yes

3. Has the statistical analysis been performed appropriately and rigorously? 

Reviewer #1: N/A

4. Have the authors made all data underlying the findings in their manuscript fully available?

Reviewer #1: Yes

5. Is the manuscript presented in an intelligible fashion and written in standard English?

Reviewer #1: Yes

6. Review Comments to the Author

Reviewer #1: The authors have adequately revised the paper following the reviewers’ suggestions and there are only two remaining aspects that in my view should be addressed in a possible revision.

(1) Although papers on other neurological diseases were excluded after the reviewers’ comments, there are still some instances that refer to these diseases (e.g. at the end of the “Scope of the review” section or the “Inclusion criteria” listed in table 3).

(2) The flow chart (figure 1) contains some errors: On the left side, the third box from above should read “Records after duplicates removed (n=4003)”, instead of “before”. In addition, the arrows pointing down on the right side should be removed (apart from the top one).

7. PLOS authors have the option to publish the peer review history of their article (what does this mean?). If published, this will include your full peer review and any attached files.

Reviewer #1: No

---

## [Author Response · Author response to Decision Letter 1]

1 Oct 2020

Reviewer comment - Although papers on other neurological diseases were excluded after the reviewers’ comments, there are still some instances that refer to these diseases (e.g. at the end of the “Scope of the review” section or the “Inclusion criteria” listed in table 3).

Our response - We have been through the paper thoroughly and removed mention of other neurological diseases. All these instances are now evident in the track changed manuscript.

Reviewer comment - The flow chart (figure 1) contains some errors: On the left side, the third box from above should read “Records after duplicates removed (n=4003)”, instead of “before”. In addition, the arrows pointing down on the right side should be removed (apart from the top one)

Our response - We have now corrected Figure 1 (flow chart) and also removed the arrows as suggested.

---

## [Editor Report · Decision Letter 2]

5 Oct 2020

A realist review of advance care planning for people with multiple sclerosis and their families

PONE-D-20-09582R2

Dear Dr. Koffman,

We’re pleased to inform you that your manuscript has been judged scientifically suitable for publication and will be formally accepted for publication once it meets all outstanding technical requirements.

Kind regards,

Andrew Soundy

Academic Editor

PLOS ONE
---

## [Editor Report · Acceptance letter]

7 Oct 2020

PONE-D-20-09582R2 

A realist review of advance care planning for people with multiple sclerosis and their families 

Dear Dr. Koffman:

I'm pleased to inform you that your manuscript has been deemed suitable for publication in PLOS ONE. Congratulations! Your manuscript is now with our production department. 

Kind regards, 

on behalf of

Dr. Andrew Soundy 

Academic Editor

PLOS ONE